# Spatiotemporal Dynamic Evolution and Its Driving Mechanism of Carbon Emissions in Hunan Province in the Last 20 Years

**DOI:** 10.3390/ijerph20043062

**Published:** 2023-02-09

**Authors:** Huangling Gu, Yan Liu, Hao Xia, Xiao Tan, Yanjia Zeng, Xianchao Zhao

**Affiliations:** 1School of Metallurgy and Environment, Central South University, Changsha 410083, China; 2School of City and Environment, Hunan University of Technology, Zhuzhou 412007, China; 3Hunan Provincial Key Laboratory of Safe Discharge and Resource Utilization of Urban Water, Hunan University of Technology, Zhuzhou 412007, China

**Keywords:** carbon emissions, LISA time path, spatiotemporal transition, migration of gravity center, GTWR model, Hunan Province

## Abstract

Global warming caused by carbon emissions is an environmental issue of great concern to all sectors. Dynamic monitoring of the spatiotemporal evolution of urban carbon emissions is an important link to achieve the regional “double carbon” goal. Using 14 cities (prefectures) in Hunan Province as an example, based on the data of carbon emissions generated by land use and human production and life, and on the basis of estimating the carbon emissions in Hunan Province from 2000 to 2020 using the carbon emission coefficient method, this paper uses the Exploratory Spatial–Temporal Data Analysis (ESTDA) framework to analyze the dynamic characteristics of the spatiotemporal pattern of carbon emissions in Hunan Province from 2000 to 2020 through the Local Indicators of Spatial Association (LISA) time path, spatiotemporal transition, and the standard deviation ellipse model. The driving mechanism and spatiotemporal heterogeneity of urban carbon emissions were studied by using the geographically and temporally weighted regression model (GTWR). The results showed that: (1) In the last 20 years, the urban carbon emissions of Hunan Province have had a significant positive spatial correlation, and the spatial convergence shows a trend of first increasing and then decreasing. Therefore, priority should be given to this relevance when formulating carbon emission reduction policies in the future. (2) The center of carbon emission has been distributed between 112°15′57″~112°25′43″ E and 27°43′13″~27°49′21″ N, and the center of gravity has shifted to the southwest. The spatial distribution has changed from the “northwest–southeast” pattern to the “north–south” pattern. Cities in western and southern Hunan are the key areas of carbon emission reduction in the future. (3) Based on LISA analysis results, urban carbon emissions of Hunan from 2000 to 2020 have a strong path dependence in spatial distribution, the local spatial structure has strong stability and integration, and the carbon emissions of each city are affected by the neighborhood space. It is necessary to give full play to the synergistic emission reduction effect among regions and avoid the closure of inter-city emission reduction policies. (4) Economic development level and ecological environment have negative impacts on carbon emissions, and the population, industrial structure, technological progress, per capita energy consumption, and land use have a positive impact on carbon emissions. The regression coefficients are heterogeneous in time and space. The actual situation of each region should be fully considered to formulate differentiated emission reduction policies. The research results can provide reference for the green and low-carbon sustainable development of Hunan Province and the formulation of differentiated emission reduction policies, and provide reference for other similar cities in central China.

## 1. Introduction

The issue of climate warming caused by a large amount of CO_2_ emissions has become the focus of global attention, which poses a great challenge to the sustainable development of nature and human society [1,2]. Slowing down the emission of greenhouse gases such as CO_2_ and promoting the development of a low-carbon economy are the common ways for all countries in the world to solve climate warming [3]. Since the reform and opening-up, the rapid development of industrialization and urbanization in China has led to the continuous increase in energy consumption and carbon emissions, which has seriously hindered the pace of national ecological civilization construction [4]. According to the statistics of the International Energy Agency (IEA), China’s CO_2_ emissions in 2018 were as high as 9.48 Gt, accounting for about 28.61% of the global total [5]. At present, as one of the largest carbon emitters in the world, China is facing the dual test of domestic economic development and international responsibility for carbon emission reduction, and at the United Nations General Assembly in 2020, it put forward the “double carbon” goal of achieving carbon peak by 2030 and carbon neutrality by 2060 [6]. As the most intensive and active region of human activities, cities are not only the largest driving force of the global carbon cycle, but also the main body undertaking the task of carbon emission reduction [7]. Therefore, it is particularly important to scientifically estimate urban carbon emissions and study the spatiotemporal dynamic characteristics and influencing factors on this basis to guide regional carbon emission reduction and low-carbon transformation and development.

In recent years, the issue of carbon emission has been widely discussed. Owing to different development levels, carbon emissions have significant spatial differences across regions, which is necessary to discuss from a spatial perspective [8,9,10]. Studies have shown that carbon emission has spatial spillover effects [11,12]. Li et al. (2019) found significant spatial autocorrelation and spatial agglomeration effects of carbon emission in 30 Chinese provinces during the period of 2004–2016 [13]. Zhang et al. (2020) demonstrated positive spatial autocorrelation of carbon emission intensity among 281 prefecture-level cities in China [14]. Some scholars began to investigate the temporal and spatial variation in carbon emission [15,16]. However, most of them only analyze from a temporal or spatial perspective, and few have traced the spatiotemporal evolution of carbon emission with precision [17]. The Exploratory Spatial Data Analysis (ESDA) model provides solutions to quantitively capture the dynamic changes in carbon emission from both temporal and spatial perspectives, which has been applied to various fields, such as carbon emission from agriculture and water use [18,19]. Rey proposed a space–time transition classification under the framework of ESDA, which is suitable to discuss the temporal and spatial changes in carbon emission [20]. Zhao et al. (2017) used the ESDA model to classify the spatiotemporal transition of 30 provinces in China from 1997 to 2015, which found that the spatiotemporal evolution characteristics of carbon intensity among provinces show both “agglomeration” and “differentiation” in the spatial distribution [21]. Relevant research areas using ESTDA are mostly concentrated in Beijing–Tianjin–Hebei [22], the Yangtze River Delta [23,24], central China [25] and eastern coastal cities [26,27]. There are relatively few studies on urban carbon emissions in central China, especially in provinces such as Hunan Province, where “resource saving society and environment friendly society” construction has not been reported. For the research on the influencing factors of carbon emissions, the research methods used are mainly ESDA (exploratory spatial data analysis), panel data regression analysis, time sequence analysis, LMDI (Logarithmic mean divisia index) decomposition method, geographic weighted regression (GWR) method, input–output analysis, STIRPAT model (Stochastic Impacts by Regression on Population, Affluence, and Technology), etc. [22,23,28,29,30]. Additionally, the research on the influencing factors of carbon emissions mainly includes economic development level, urbanization rate, energy consumption structure, population, industrial structure, etc. [23,24,25,26,27,28,29,30,31]. Most of the current studies on the influencing factors of carbon emissions do not consider the temporal and spatial differences in each driving factor, but only draw a conclusion based on panel data regression, which cannot reflect the temporal and spatial differences in the influencing factors of carbon emission in China. However, the geographically and temporally weighted regression model (GTWR) can study the spatiotemporal heterogeneity of each driving factor [32,33]. 

In view of this, this study takes 14 cities (prefectures) in Hunan Province as an example, based on the data of carbon emissions generated by land use and human activities, and uses the carbon emission coefficient method to estimate the carbon emissions of 14 cities from 2000 to 2020. Then, using the ESTDA framework, through the methods of LISA time path, spatiotemporal transition, and standard deviation ellipse model (The ESTDA method is described in the Appendix A), from the perspective of spatiotemporal interaction, the spatiotemporal evolution characteristics and gravity center migration of urban carbon emissions in the last 20 years are discussed. Moreover, the GTWR model is introduced to explore the spatiotemporal heterogeneity of the impact degree of various driving factors of carbon emissions (The GTWR method is described in the Appendix A), in order to more clearly reveal the driving mechanism of the spatial differentiation of carbon emissions in each city in Hunan Province, which is of great significance to deeply understand the spatiotemporal patterns of carbon emissions in central cities and formulate differentiated emission reduction policies.

## 2. Methodology and Data

### 2.1. Overview of the Study Region

Hunan Province is in the middle of China, where the middle reaches of the Yangtze River are surrounded by Jiangxi, Chongqing, Guizhou, Guangdong, Guangxi, and Hubei, including Changsha, Zhuzhou, Xiangtan, Hengyang, Yueyang, Loudi, Shaoyang, Changde, Yiyang, Chenzhou, Yongzhou, Huaihua, and Xiangxi Tujiazu and Miaozu Autonomous Prefecture (Figure 1). With a land area of 211,800 km^2^, the total population of Hunan Province was 66,444,000 in 2020. The province’s total carbon emissions reached 3.1 × 10^8^ t in 2019. Most of Hunan Province belongs to terrestrial ecosystems and has abundant natural resources, such as forests, grasslands, and wetlands, playing a very important role in carbon cycling and reducing atmospheric concentration.

### 2.2. Data Sources

This study collected and preprocessed the data of land use and carbon emission of 14 cities in Hunan Province. The land use data come from the Resource and Environmental Science and Data Center of the Chinese Academy of Sciences (http://www.resdc.cn, accessed on 20 August 2022), with a resolution of 30 m × 30 m, including 6 first-level land types such as cropland and forest, and 24 second-level land categories such as paddy field and dry land. After reclassification, the six categories cropland, forest, grassland, water, construction land, and unused land were obtained, and five periods of land use data in 2000, 2005, 2010, 2015 and 2020 were obtained. The different types of land and its utilization in Hunan Province from 2000 to 2020 are shown in Figure 1. The carbon emission data are based on the emission factor multiplied by the activity level. The carbon emission factors include fossil energy combustion, power consumption, industrial production process, human, livestock respiration, soil respiration, and the data of various carbon emission factors were derived from the statistical yearbook of Hunan Province, China energy statistical yearbook, Hunan energy statistical yearbook and Hunan energy development report from 2000 to 2020. The vector administrative boundary data is from the National Geomatics Center of China (http://www.ngcc.cn, accessed on 15 June 2022).

### 2.3. Calculation Method of Carbon Emissions


(1)Carbon emissions from land use


Based on the distribution and size of different vegetation types in cities in Hunan Province, combined with research on different carbon emission vegetation in China, the main carbon emission vegetation in this study area were identified within cropland, forest, grassland, water, construction land, and unused land. Among them, carbon emission was found in construction land and cropland, and carbon absorption was found in forest, grassland, water and unused land. The calculation is shown in Equation (1).
(1)$$E_{\mathrm{cs}}=\sum e_{i}=\sum A_{i}\times \beta_{i}$$
where *E_cs_* is the total carbon emissions from land use; *i* is the *i*-th land type; *e_i_* is the carbon emission of the *i*-th land type; *A_i_* is the area of the *i*-th land type; *β_i_* is the carbon emission coefficient of the *i*-th land type (carbon emission is positive, carbon absorption is negative); and the value is determined by referring to the existing research results (Table 1).

The carbon emission of cropland needs to be considered in two aspects: one is that agricultural production and irrigation processes produce a large amount of greenhouse gases such as CO_2_ and CH_4_, and the other is that crops can absorb a certain amount of CO_2_ through photosynthesis during the growth period [34]. According to the research of Cai et al. [35] and He et al. [36], the carbon emission coefficient of cropland was 0.504 t·hm^−2^, and the carbon absorption coefficient was 0.007 t·hm^−2^. Therefore, this study used the difference between the two to determine the carbon emission coefficient of cropland to be 0.497 t·hm^−2^. According to the research results of Shi et al. [37] and Xiao et al. [38], the weighted average value of the vegetation carbon sink efficiency of forest in China was −0.581 t·hm^−2^. Therefore, the carbon emission coefficient of forest in this study was determined to be −0.581 t·hm^−2^. According to Fang et al. [39], the grassland carbon emission coefficient was −0.021 t·hm^−2^, and Li et al. [40] set the grassland emission coefficient in Wuhan to be −0.022 t·hm^−2^, which has a small range of change. Therefore, this study determined the grassland carbon emission coefficient to be −0.021 t·hm^−2^. In general, water is considered a carbon sink. According to Lai et al. [41] and Duan et al. [42], the carbon emission coefficients of waters were determined to be −0.257 t·hm^−2^ and −0.248 t·hm^−2^, respectively. Therefore, this study selected the carbon emission coefficient of water to be −0.253 t·hm^−2^ by the average of the above research results. The unused land area of each city and prefectures in Hunan Province is small, and it is mainly swampland and bare rock gravel land, with weak carbon absorption capacity. Referring to the research of Wei et al. [27], Lai et al. [41], Fan et al. [43], and Liu et al. [44], and taking the average value of the above research results [27,40,43,44], this study selected −0.005 t·hm^−2^ to be the carbon emission coefficient of unused land.


(2)Carbon emissions from construction land


The carbon emissions of construction land are measured through the carbon emissions generated by human production and life, specifically including carbon emissions of energy consumption, industrial production, and human and livestock respiration [45]. This study considers the actual situation of energy consumption in Hunan Province. In order to reduce the negative impact of statistical errors on the research results, the selection of energy types should be as comprehensive as possible, including raw coal, cleaned coal, other washed coal, coke, coke oven gas, blast furnace gas, other gas, other coking products, crude oil, gasoline, kerosene, diesel oil, fuel oil, liquefied petroleum gas, other petroleum products, natural gas, heat and electricity, and the conversion coefficient and corresponding carbon emission coefficient of various energy standard coals refer to the data in the literature [46,47] (Table 2). The formula for calculating carbon emissions from energy consumption is shown in Equation (2).
(2)$$E_{cn}=\sum_{i=1}^{n}m_{i}\cdot n_{i}\cdot a_{i}$$
where *E_cn_* is the regional carbon emissions from energy consumption; *m_i_* is the final consumption of the *i*-th energy; *n_i_* is the standard coal conversion coefficient of the *i*-th energy; and *a_i_* is the carbon emission coefficient of the *i*-th energy.

The carbon sources in the industrial production process mainly come from the mining industry, chemical industry, metal industry, non-energy products used in fuels and solvents, electronics industry, fluorinated substitutes for ozone-depleting substances, etc. Among them, the CO_2_ emissions in the production process of cement, steel, pig iron, calcium carbide, etc., account for more than 90% of the greenhouse gas emissions in industrial production processes [48]. Therefore, only the carbon emissions from industrial processes such as cement, steel, pig iron, and calcium carbide are considered here. In addition, cement clinker accounted for 65% of the total cement [49]. The calculation is shown in Equation (3).
(3)$$E_{ci}=\sum_{m=1}^{m=4}\left(Q_{im}\times F_{im}\right)$$
where *E_ci_* is the carbon emissions from industrial processes; *Q_im_* is the industrial production; and *F_im_* is the carbon emission coefficient of the corresponding industrial product. The carbon emission coefficient of cement, steel, pig iron, and calcium carbide are 0.0954 t/a, 0.0025 t/a, 0.0410 t/a, and 0.3147 t/a, respectively [50].

The calculation of carbon emissions from human and livestock respiration is shown in Equation (4).
(4)$$E_{cb}=Q_{p}\times C_{p}+\sum_{t=1}^{t=2}\left(Q_{at}\times C_{at}\right)$$
where *E_cb_* is the total carbon emissions from both human and livestock respiration; *Q_p_* is the resident population in each city in Hunan; *C_P_* is the carbon dioxide emission coefficient of human respiration; and *Q_at_* is the number of animals at the end of stocks in each city in Hunan. Considering that small livestock have little effect on carbon emissions, this paper only considered cattle and pigs, as they are the main large livestock in Hunan Province. *C_at_* is the carbon dioxide emission coefficient of the corresponding livestock, and the carbon dioxide emission coefficients of human, cattle, and pigs were 0.079 t/a, 0.796 t/a, and 0.082 t/a, respectively [51].


(3)Carbon emission of each city


Calculation formula:*E_cr_* = *E_cs_* + *E_cn_* + *E_ci_* + *E_cb_*(5)
where *E_cr_* is the carbon emissions of *i*-th city in Hunan Province; *E_cs_*, *E_cn_*, *E_ci_*, *E_cb_* have the same meaning as above.

## 3. Results and ESTDA Analysis

### 3.1. Time Characteristics of Carbon Emission

Based on the land use and energy consumption data of Hunan Province from 2000 to 2020, the total carbon emissions of each city from 2000 to 2020 were obtained (Figure 2). From 2000 to 2010, the total carbon emissions in Hunan Province continued to increase, from 25.03 Mt in 2000 to 10,056.02 Mt in 2010, an increase of nearly five times. Due to the rapid growth in investment during this period, high energy-consuming industries such as electric power and thermal production, cement, and steel expanded rapidly, resulting in rapid growth in energy consumption. In addition, the rapid growth in residents’ living energy consumption accelerated the growth in energy consumption in the whole society, so the growth rate of carbon emissions was larger than that in previous years. From 2010 to 2020, the total carbon emissions in Hunan Province continued to decrease, from 10,056.02 Mt in 2010 to 8676.91 Mt in 2020, mainly because of the formulation of carbon emission reduction policy, the improvement in clean energy technology, and the transformation and upgrading of industrial structures during this period, which largely contributed to the decline in energy intensity. According to the coefficient of variation calculated by each city’s carbon emissions over the years, the coefficient of variation fluctuates and rises in general. Due to the characteristics of each city’s natural resources, as well as the differences in urbanization process, population, and economic level among cities, the regional carbon emissions showed significant differences.

### 3.2. Global Spatial Correlation of Carbon Emissions

The global Moran’s *I* index of carbon emissions in Hunan Province in 2000, 2005, 2010, 2015, and 2020 calculated by GeoDa software 1.20 were 0.167, 0.172, 0.175, 0.059, and 0.131, respectively, and the results were all positive and passed the significance test (*p* < 0.05). This shows that the spatiotemporal distribution of carbon emissions between cities in Hunan Province is not completely random, but presents a significant positive spatial correlation; that is, the cities with high (low) carbon emissions tend to gather near high (low) values in space. From the change trend of the global Moran’s *I* index with time, the overall trend from 2000 to 2010 was upward, rising from 0.167 in 2000 to 0.175 in 2010, indicating that the spatial convergence of urban carbon emissions gradually increased during this period. However, there was a downward trend from 2010 to 2020, indicating that the spatial convergence of urban carbon emissions in Hunan Province continued to weaken and the spatial difference was gradually expanding during this period.

### 3.3. The Migration of Gravity Center and SDE Analysis of Carbon Emissions

In order to reveal the migration and change in the carbon emission center of gravity and the spatial distribution of high carbon emission areas in Hunan Province, the SDE method was used for complementary analysis. As can be seen from Table 3 and Figure 3, in the last 20 years, the carbon emissions gravity center in Hunan Province was distributed between 112°15′57″~112°25′43″ E and 27°43′13″~27°49′21″ N, located in the northeast of the geometric center of the whole region (111°42′31″ E, 27°36′33″ N), indicating that the carbon emissions in the eastern and northern regions of Hunan Province are relatively high. This is because the economic development level in the eastern region is relatively high in terms of economic level and social consumption factor. The rapid economic development is accompanied by a large amount of energy consumption; At the same time, the growth in social consumption in the eastern region would drive the expansion of energy demand, thus promoting an increase in carbon emissions from urban residents’ living energy. From the perspective of industrial structure and resource endowment factors, the proportion of secondary industry in northern Hunan is large, leading to higher carbon emissions in northern Hunan. Therefore, when formulating differentiated emission reduction policies, both northern and eastern cities should undertake more emission reduction tasks.

From the moving track of the center of gravity, the center of gravity of carbon emissions in Hunan Province moved to the southwest overall, indicating that the growth rate of carbon emissions in the western and southern cities of Hunan Province is higher than the average level. In the last 20 years, the moving distance was 49.44 km in total, which is reflected in that the center of gravity migrated 9.55km to the northeast from 2000 to 2005, 16.85 km to the southwest from 2005 to 2010, 11.99 km to the northwest from 2010 to 2015 and 11.05 km to the northeast from 2015 to 2020. According to the change in the azimuth angle, in the last 20 years, there has been a trend of decreasing fluctuations on the whole, first increasing from 173.39° in 2000 to 175.96° in 2005, then continuously decreasing to 0.23° in 2015, and finally increasing to 3.74° in 2020, indicating that the spatial pattern of carbon emissions in Hunan Province presented a northwest–southeast pattern in the last 20 years, but this pattern has been weakened as a whole and gradually changed to the north–south direction.

From the main axis, the long axis increased from 143.01 km in 2000 to 146.50 km in 2005, indicating that the carbon emissions in Hunan Province were scattered in the “northwest–southeast” direction during this period, mainly due to the significant increase in carbon emissions in old industrial cities such as Zhuzhou and Xiangtan, while a slow growth in carbon emissions was seen in tourist cities such as Xiangxi and Zhangjiajie. From 2005 to 2020, the long axis decreased from 146.50 km to 136.58 km, showing that the carbon emissions appeared in a centripetal agglomeration in the “northwest–southeast” direction during this period. This is because economic development, as one of the leading factors of carbon emission growth, was weak in the economic foundation of southern and western Hunan. At the beginning of the implementation of economic development policies, such as the development of the western Hunan Province and the strengthening of provincial councils, the development speed was slow, and the carbon emission was low, while the economic development of Changsha, Zhuzhou, and Xiangtan was rapid, and higher carbon emissions promoted the centripetal agglomeration of carbon emission.

From the auxiliary axis, the minor axis had decreased from 199.24 in 2000 to 184.39 km in 2020, indicating that the carbon emissions in Hunan Province have been polarized from northeast to southwest in the last 20 years. The main reasons for this are: on the one hand, from the perspective of economic development, the rapid economic development of eastern and northern Hunan in recent years has led to a rapid growth in carbon emissions in northeastern cities; on the other hand, from the perspective of industrial structure, Chenzhou, an old industrial base, is highly dependent on the secondary industry. In recent years, it has continuously attracted investment and a large number of heavy industry enterprises, resulting in a large demand for energy consumption, which makes its carbon emissions show a divergent trend.

### 3.4. Analysis of LISA Time Path

According to the temporal variation characteristics of the global Moran’s *I* index, the carbon emissions from 2000 to 2010 were in a continuous rising stage, while they were then in a continuous declining stage from 2010 to 2020 (Figure 2). Therefore, 2010 was taken as a node, and the two periods 2000–2010 and 2010–2020 were selected for LISA time path analysis.

#### 3.4.1. Geometric Feature Analysis

The dynamics of local spatial structure of carbon emissions in Hunan Province and the fluctuation in spatial dependence direction can be revealed by the geometric characteristics of the LISA time path [52]. In this study, the relative length of the LISA time path from low to high in 2000–2010 was divided into four grades, namely, low relative length (0.155~0.355), medium relative length (0.356~0.747), high relative length (0.748~1.215), and maximum relative length (1.216~2.812), by using a natural breakpoint classification method (Jenks) and manual classification method. The relative length of the LISA time path in 2010–2020 was classified by the manual classification method too, and the classification level and interval range were consistent with those in 2000–2010. Figure 4a shows that from 2000 to 2010, Changsha, Yueyang, and Loudi had high relative lengths, indicating that their local spatial structure is highly dynamic. The cities with higher relative lengths include Zhuzhou, Xiangtan, and Changde, which shows that the local spatial structure of these three cities has a strong dynamic character. Yiyang, Huaihua, and Zhangjiajie had low relative lengths, indicating that their local spatial structures have the strongest stability. Other cities were of medium relative lengths, indicating that they belong to relatively stable local spatial structures. In 2010–2020, the number of cities with high relative lengths and high relative lengths increased significantly compared with those in 2000–2010 (Figure 4b), indicating that the number of cities with strong volatility is on the rise, and these cities are mainly distributed in western and southern Hunan. The reason is that the implementation of the “Eastern Industry Transfer” policy in Hunan Province guarantees its economic development, but also brings many environmental problems. Among them, Xiangtan and Hengyang, old industrial bases, are dominated by the secondary industry with high carbon emissions. Due to the influence of policies, the local spatial structure of these cities has a strong dynamic. There are five cities with medium and low relative lengths showing a downward trend, because these cities have changed from medium and low relative lengths to high relative lengths, such as Zhangjiajie, Huaihua, Shaoyang, Chenzhou, and Xiangxi. On the one hand, the rapid economic development in these regions in recent years has led to a significant improvement in residents’ living standards and social consumption levels, and consumption has become increasingly rich, from the survival type to the luxury type [53]. On the other hand, the rapid development of industries such as real estate and the automobile industry has driven the rapid expansion of high-carbon manufacturing industries such as the mining, oil and metal processing, and building materials industries, which has led to a significant increase in carbon emissions and caused the local spatial structure of regional carbon emissions to become more dynamic. From 2000 to 2020, there were nine cities with a relative length less than the average length, accounting for 64%, which shows that the local spatial structure of urban carbon emissions in Hunan Province has had a strong stability in the last 20 years.

Similarly, the LISA time path curvature from low to high in 2000–2010 was divided into four grades by the natural breakpoint method: low curvature (1.000~1.692), medium curvature (1.693~2.718), high curvature (2.719~4.958) and maximum curvature (4.959~7.404). The curvature of the LISA time path from 2010 to 2020 was graded by the same classification method. As shown in Figure 5, from 2000 to 2010, Zhuzhou was a city with high curvature, indicating that Zhuzhou had a highly volatile carbon emission growth and local spatial dependence change process. Most of the urban areas in central Hunan, southern Hunan, and eastern Hunan belonged to low curvature cities, with weak volatility of carbon emission growth and local spatial dependence change process. Other urban areas had moderate curvature. From 2010 to 2020, Xiangxi became the urban area with the largest curvature, while Changde, Loudi, and Xiangtan had higher curvature, and the urban areas with the strongest volatility in the direction of spatial dependence showed an increasing trend from one in 2000 to four in 2010. The number of urban areas with low curvature showed a downward trend; that is, the number of urban areas with weak volatility of carbon emission growth and local spatial dependence change process decreased. To sum up, the proportion of cities with low curvature and medium curvature in the two periods was more than 70%, indicating that the carbon emission growth and local spatial dependence change process in Hunan Province are relatively stable as a whole. From the perspective of spatial distribution, the curvature of most urban areas in northern Hunan, western Hunan, and central Hunan showed an increasing trend; that is, the volatility in the direction of spatial dependence increased, while the curvature grade in southern Hunan was relatively stable; that is, the volatility in the direction of spatial dependence tended to be relatively stable, because the impact of industrial carbon emissions in these areas is small, making the local spatial dependence change process of urban carbon emissions relatively stable.

#### 3.4.2. Average Moving Direction Analysis

The LISA moving direction can analyze the spatial integration of carbon emission spatial pattern changes in Hunan Province. According to the LISA coordinates of each city at two time points, the LISA moving direction can be divided into four types based on the average level: 0°~90° direction (high–high trend), indicating that the carbon emissions of the city itself and its adjacent urban areas maintain a high growth trend; 90°~180° direction (low–high trend), indicating that the carbon emissions of cities show a low growth trend, while the adjacent cities show a high growth trend; 180°~270° direction (low–low trend), meaning the carbon emissions of the city itself and its adjacent cities show a low growth trend; and 270°~360° direction (high–low trend), indicating that the carbon emissions of cities is in a high growth trend, while that of adjacent cities is in a low growth trend. Among them, the 0°~90° direction is a synergistic high growth trend, and the 180°~270° direction is a synergistic low growth trend. These two synergistic growth trends represent the integrated spatial dynamics of cities and their adjacent cities.

It can be seen in Figure 6 that from 2000 to 2010, there were three cities with coordinated high growth, namely Hengyang, Huaihua, and Xiangxi. There were three cities with coordinated low growth, namely Chenzhou, Yongzhou, and Zhangjiajie. It shows that the spatial pattern evolution of carbon emissions in Hunan Province had a weak spatial integration in the period of 2000–2010. However, from 2010 to 2020, only Yiyang and Xiangxi had coordinated high growth. The reason is that with the adjustment of industrial layout in Hunan Province in 2010, the tourism industry in Xiangxi developed rapidly, and the secondary industry in Yiyang continued to develop, resulting in a rapid increase in carbon emissions. There were five cities with coordinated low growth, namely Zhuzhou, Xiangtan, Loudi, Hengyang, and Chenzhou. Comparing the two periods, we concluded that the number of urban areas with coordinated high and low growth has increased from six in 2000–2010 to seven in 2010–2020, indicating that the spatial pattern change in carbon emissions in Hunan Province has a certain spatial integration, and this spatial integration shows an increasing trend.

### 3.5. Analysis of LISA Spatiotemporal Transition

The LISA time path analysis only revealed the changing size and trend of LISA coordinates in each city; however, it cannot reflect the mutual transfer characteristics of local spatial correlation types in Moran’s *I* scatterplot. Subsequently, Rey [54] proposed Moran’s *I* transition probability matrix and spatiotemporal transition to effectively solve this problem. Therefore, this study uses the probability transfer matrix and spatiotemporal transition proposed by Rey to explore the transfer characteristics and evolution process of local spatial correlation types of carbon emissions in Hunan Province.

It can be seen from Table 4 that the probability of Moran’s *I* scatter remaining in the same quadrant (type I) in the four periods of 2000–2005, 2005–2010, 2010–2015, and 2015–2020 is more than 50%, indicating that there is a certain transfer inertia in carbon emissions in Hunan Province, and the type is not easy to change, with strong path dependence and spatial locking characteristics. The proportion of types II and III in four periods accounts for about 7%, which indicates that there is a certain transition probability between local spatiotemporal correlation categories of carbon emissions in Hunan Province. The proportion of type IV in the four periods is 0%, showing that the probability of jump transfer of carbon emissions in Hunan Province is extremely low.

According to the number of various types of transitions, the spatiotemporal cohesion index (SC) of the four periods is above 50%, and the spatiotemporal transition index (SF) is mostly less than 30%, which further shows that the local spatial correlation model of carbon emissions in Hunan Province has strong stability; that is, it has certain path dependence or spatial locking characteristics. With the passage of time, the SC index shows an upward trend, while the SF index shows a downward trend, indicating that the path dependence and locking characteristics of the spatial pattern of carbon emissions in Hunan Province are gradually enhanced.

## 4. Analysis of Influencing Factors

### 4.1. Data Verification

Using the GTWR model, the carbon emissions in Hunan Province were taken as the dependent variables, according to the research of other scholars on the driving factors of carbon emissions [23,25,29,30,33,55]. The explanatory variables affecting carbon emissions include population, economic development level, industrial structure, technological progress, energy intensity, land use, and ecological environment (Table 5). Before making the GTWR model, all variables need to be standardized. In order to avoid the pseudo-regression phenomenon during regression, a multicollinearity test was carried out on all standardized variables by the regression analysis method, and the variance inflation factor (VIF) of all variables was less than 10 (Table 5). Therefore, the above seven indicators were used as explanatory variables in the GTWR model. Table 6 shows the related parameters of the GTWR regression results. From the goodness of fit, both R^2^ and corrected R^2^ were higher than 0.96, indicating that the GTWR regression model can better measure the influence of explanatory variables on dependent variables.

### 4.2. Spatiotemporal Heterogeneity of Carbon Emission Influencing Factors

In order to observe the spatiotemporal distribution difference in regression coefficients of various influencing factors more intuitively, the regression coefficients were averaged and visualized. The results are shown in Figure 7. More generally, there were spatiotemporal differences in the regression coefficients of various influencing factors. 

(1) The influence of population (POP) on carbon emissions gradually weakened. There was a positive correlation between population and carbon emissions in general, and the regression coefficient span was small, distributed between 0.259 and 0.358. High-value areas were concentrated in eastern Hunan and southern Hunan from 2000 to 2010, and the number of cities in high-value areas decreased from 2010 to 2020. The regions with low regression coefficients were mainly in western Hunan, where the influence of population factors on carbon emissions was weak. The influence degree of population on carbon emissions in Hunan Province showed a decreasing trend from southeast to northwest, in which the high-value regions were mainly located in eastern Hunan, and the low-value regions were mainly in western Hunan. Most of the population in Hunan Province was concentrated in the east and north. Population agglomeration can bring about spatial agglomeration of economic activities and production factors, and the increase in population will also bring about an increase in energy demand, which will lead to an increase in carbon emissions. Compared with other regions, the level of urbanization in western Hunan is low and the economic development is weak, which leads to more and more people going to eastern and northern Hunan for employment. As a result, the impact of population factors in western Hunan on carbon emissions is lower than that in northern Hunan.

(2) The contribution of economic development level (PGDP) to carbon emissions fluctuates greatly, from a negative impact in 2000–2010 to a positive impact in 2010–2020. From 2000 to 2020, the absolute value of the regression coefficient showed a decreasing trend, indicating that the impact of economic development level on carbon emissions gradually weakened. As the improvement in economic development level brought about the transformation of industrial structure, the economy began to develop in an intensive direction, and the efficiency of resource utilization continued to improve, thus causing the carbon emission intensity continue to decline. Secondly, with the development of the economy, people’s awareness of environmental protection is constantly improving, and they begin to pursue a low-carbon lifestyle, which also brings about a decline in carbon emissions. With the continuous optimization of economic development mode and the implementation of energy conservation and emission reduction policies, the impact of economic development on carbon emissions will be further reduced in the future. In the last 20 years, Hunan Province has made some achievements in promoting the optimization and upgrading of the industrial structure and the transformation of the economic development model. While the economic level has improved, it has also begun to pay attention to environmental issues, and the technical level has also been continuously improved, resulting in a reduction in carbon emissions. For Hunan Province, as a developing province, how to control and reduce carbon emissions and energy consumption while maintaining stable and rapid economic development, as well as further enhance the technical support capacity of energy conservation and emission reduction, is a realistic problem that needs to be considered and solved for a long time in the future.

(3) The influence of industrial structure (IS) on carbon emissions tends to be stable, both of which are positive, with obvious intensity. Industrial development is highly dependent on energy, so the increase in industrialization rate will inevitably lead to an increase in energy consumption, thus increasing carbon emissions. With the continuous progress of technology, the resource utilization efficiency of enterprises improves, while energy consumption decreases, and the gap between industrialization levels in different regions narrows, the technology spillover effect between regions increases, the compact of industrialization level on carbon emissions gradually weakens, and the dispersion degree of regression coefficient shrinks. Chenzhou, Yueyang, and Zhuzhou had a greater impact on industrial structure, followed by Yongzhou and Changsha, and Huaihua, Zhangjiajie, and Xiangxi had a weaker influence. The development of industries varies among cities in Hunan Province, with the tertiary industry accounting for the highest proportion, and tourism being relatively developed in Xiangxi. Yiyang and Changde had the same proportion of secondary and tertiary industries, mainly light manufacturing and tourism, and their industrial structures were relatively low-carbon and environmentally friendly. Chenzhou, Yueyang, and Zhuzhou were all dominated by traditional industries with high energy consumption and high emission, such as traditional iron and steel, mining, dressing, and metallurgy, and their industrial structure needs to be optimized urgently.

(4) The impact of technological progress (EI) on carbon emissions is becoming increasingly evident. The positive influence is dominant, with obvious spatial differentiation. The positive high-value area is gradually concentrated from southern Hunan to eastern Hunan, and the regression coefficient changes more and more with time, which indicates that technological progress is increasingly powerful in explaining carbon emissions. Low-carbon production technology can improve energy utilization efficiency to reduce energy consumption and pollution emissions per unit of GDP. At the same time, with the accelerated urbanization process, the transformation and upgrading of industrial structure, the development focus has gradually shifted from industry to modern service industry and high-tech industry, which has greatly contributed to the reduction in energy intensity. Technological progress had an aggravating effect on carbon emissions in Hunan Province, and the coefficients of technological progress were positive, which showed that technological progress can stimulate economic growth. The improvement in technical and management level improved the energy utilization efficiency, thus reducing carbon emissions. Therefore, it played a positive role in affecting carbon emissions. The regression coefficient gradually increased from the northwest to the southeast, which showed that technological progress has the greatest impact on carbon emissions in eastern and southern Hunan. Analysis of the reasons shows that eastern and southern Hunan has rapid economic development, rapid industrialization, and urbanization, and a large proportion of the secondary industry, both in terms of technical and management levels, is in the leading position in Hunan Province. Therefore, the technological progress factor has the most obvious influence on carbon emissions in eastern and southern Hunan. In western Hunan, the level of economic development is low, with a high proportion of primary industry and low energy consumption, so the impact of technological progress on carbon emissions in western cities is also small. From the size of the regression coefficient of the overall technological progress factors, the regression coefficient gradually increased from 2000 to 2020, indicating that the overall industrial technology and management level have been improved to a certain level. At the same time, this shows that the reduction in energy efficiency alone is not enough to achieve the goal of emission reduction, and it also needs to adjust the industrial structure and change the mode of economic development.

(5) The influence of per capita energy consumption (PEC) on carbon emissions was relatively volatile, but it was generally shown to be a decreasing trend over the last 20 years. The regression coefficient changes with time were smaller and smaller, indicating that the explanatory power of PEC was becoming weaker and weaker. Due to the continuous progress of urbanization, population was constantly gathering in cities. Population gathering guides economic activities and production factors to gather in space and share infrastructure by exerting the cost optimization effect, thus improving the comprehensive utilization efficiency of energy and resources, and at the same time saving the cost of emission reduction to the maximum extent, which ultimately has less and less impact on carbon emissions. The PEC coefficient was positive, and the regression coefficient showed a pattern of high in northwest and low in southeast, which indicates that the urbanization level in western and northern Hunan Province was relatively low, the cities were still expanding extensively, and the urban infrastructure construction had brought about an increase in energy consumption and carbon emissions. However, due to the continuous progress of urbanization in the eastern and southern regions, population was constantly gathering in cities. Population gathering can improve the comprehensive utilization efficiency of energy and resources by exerting the cost optimization effect and sharing infrastructure, and finally had less and less impact on carbon emissions.

(6) The impact of land use (CON) on carbon emissions fluctuates greatly, from a negative impact in 2000–2010 to a positive impact in 2010–2020, and the regression coefficient shows a trend of increasing fluctuation. Because of the rapid development of urbanization in the later period of the study, the number of built-up areas has been increasing, which has led to the vigorous development of the real estate industry and construction industry, and the rapid increase in the urban construction area and the extensive and inefficient urban expansion mode have led to a rise in carbon emissions. The land use coefficient of Yongzhou was the largest, followed by Chenzhou, Hengyang, and Shaoyang. This showed that there was a positive correlation between land use factors and carbon emissions in these cities, mainly because the urban development mode in this region was relatively extensive, and the problems such as over-development of real estate, urban “pie-spreading” expansion, and inefficient land use have caused great waste of resources and energy consumption, resulting in an increase in carbon emissions.

(7) Because of the carbon sink effect of the ecosystem, the impact of ecological environment (ES) on carbon emissions is negative. The absolute value of regression coefficient gradually decreased from 2000 to 2020, and the number of cities with a negative impact on high-value areas significantly decreases. Due to the rapid development of urbanization, extensive urban expansion did not consider the optimization of land space function zoning and land use structure, and the boundary of urban sprawl was not reasonably determined. The forest coverage rate gradually reduced, and the ecological carrying capacity became worse and worse, leading to the gradual weakening of the impact of ecological environment on carbon emissions. The absolute value of regression coefficient increased from southwest Hunan to northeast Hunan. The rapid development of urbanization in northeast Hunan will inevitably lead to a reduction in the area of cropland and forest and the expansion of the scale of construction land along with the urbanization process. In addition, the continuous advancement of industrialization will also bring about a rapid increase in energy consumption, which will lead to a substantial increase in carbon emissions from construction land. The forest loss caused by the requisition and occupation of forest is serious, and the ecological carrying capacity is becoming worse and worse, leading to the negative impact of ecological environment on carbon emissions. However, western Hunan is rich in forest resources, and the construction of the Great Xiangxi Ecotourism Area has restricted the entry of high energy-consuming industries, keeping the ecological carrying capacity at a stable state, so the negative impact of the ecological environment on carbon emissions in western Hunan is weak. Therefore, strictly controlling the requisition and occupation of forest to reduce forest losses and moderately carrying out forest management to improve growth rate are important measures to enhance forest carbon fixation capacity.

The change trend in each influencing factor shows that the carbon emissions in Hunan Province are mutually affected, reflecting the interaction of urban space. At different stages of urbanization, the dominant factors and their influence areas have obvious regional differences, which has certain guiding significance for clarifying the urban function orientation and development direction.

## 5. Conclusions

The economic construction of Hunan Province has regional planning characteristics in the process of urbanization and industrial development, and is an important part of high-quality comprehensive development in central China. In this paper, the panel data and land use data of Hunan Province from 2000 to 2020 were selected, and the ESTDA analysis method and GTWR model were integrated to deeply analyze the spatiotemporal variation rules and driving factors of carbon emissions in Hunan Province, to create a positive exploration of the green and low-carbon economic development mode and the formulation of carbon emission reduction policies. The main conclusions are as follows:

(1) From 2000 to 2020, the overall Moran’s *I* index of urban carbon emissions in Hunan Province is positive, and shows a trend of first rising and then falling over time. All of them pass the 5% significance test, indicating that there is a significant positive spatial correlation between urban carbon emissions in Hunan Province; that is, cities with higher or lower carbon emissions tend to be adjacent in space. Therefore, priority should be granted to this relevance when formulating carbon emission reduction policies in the future. Under the background of regional coordinated development, we should strengthen the flow of factors, industrial cooperation, and optimization and adjustment among cities, to make urban industries development have linkage characteristics. At the same time, we should increase support for green industries, improve the mechanism of regional cooperation and mutual assistance and benefit compensation system in cities, achieve the goal of carbon emission reduction, and achieve the national “double carbon” strategic goal.

(2) In the last 20 years, the carbon emission center of gravity in Hunan Province has changed between 112°15′57″~112°25′43″ E and 27°43′13″~27°49′21″ N. The carbon emission center of gravity for each period is in the northeast of the geometric center of Hunan Province, but the overall carbon emission center of gravity tends to move to the southwest, indicating that the growth rate of carbon emission in western and southern urban areas of Hunan Province is higher than that of other urban areas. Therefore, cities in western and southern Hunan are the key areas of carbon emission reduction in the future. On the one hand, goals are to strengthen the effective monitoring and management of carbon emissions in the region, increase the constraint of carbon emissions, optimize the traditional economic development path dominated by coal and heavy industry, promote the transformation and upgrading of energy-intensive industries such as the chemical, electric power, and steel industries, and achieve energy utilization efficiency and emission reduction results. On the other hand, it is suggested to advocate for the low-carbon consumption patterns and lifestyle of residents, optimize consumption structure, improve residents’ low-carbon awareness, and form a good social atmosphere of green and environmental protection. Finally, in addition to limiting high carbon emissions, urban low-carbon development should also try to improve regional carbon sinks. Forest land and grassland are important carriers of carbon sink. It is suggested to improve the regional ecological compensation mechanism, compensation standards, and implementation systems, and effectively increase the carbon sink of the regional ecosystem through the conversion of farmland to forests and afforestation.

(3) The relative length analysis of the LISA time path shows that the number of cities with strong volatility in local spatial structure is on the rise, while the number of cities with strong stability is on the decline. In general, 64% of cities have a relative length less than the average length from 2000 to 2020, indicating that the local spatial structure of carbon emissions in Hunan Province has had a strong stability in the last 20 years. Among the moving direction types of the LISA time path, the number of urban areas with coordinated movement of carbon emissions in Hunan Province increased from six in 2000–2010 to seven in 2010–2020, which indicates that the spatial pattern change in carbon emissions in Hunan Province has a certain spatial integration, but shows an increasing trend. The LISA spatiotemporal transition analysis shows that the spatiotemporal aggregation index of four periods from 2000 to 2020 is greater than 50%, and the spatiotemporal transition index is mostly less than 30%, indicating that the local spatial correlation model of carbon emissions in Hunan Province has certain stability; that is, it has certain path dependence or spatial locking characteristics. Based on the LISA analysis results, urban carbon emissions have a strong path dependence in spatial distribution, the local spatial structure has strong stability and integration, and the carbon emissions of each city are affected by the neighborhood space. Among them, southern Hunan has relatively stable local spatial structure and spatial dependence direction, while northern, western, and central Hunan have the opposite. In addition, in the process of temporal and spatial changes in carbon intensity, there are unbalanced characteristics of emission reduction among neighboring cities, such as Yiyang and Yueyang, Zhuzhou and Xiangtan, Zhangjiajie and Xiangxi, etc. There is a certain degree of temporal and spatial competition. Therefore, to carry out low-carbon actions, it is necessary to give full play to the synergistic emission reduction effect among regions and avoid the closure of inter-city emission reduction policies.

(4) There are differences in the influence of each driving factor on carbon emissions in Hunan Province, and the same influencing factors have different effects on carbon emissions in different urban areas. With the rapid development of economy, industrialization and urbanization, and technological progresses, the spatial heterogeneity pattern of each influencing factor on carbon emissions evidently also changes. Population, industrial structure, technological progress, per capita energy consumption, and land use have a positive impact on carbon emissions, while economic development level and ecological environment have a negative impact, and each regression coefficient is spatiotemporal non-stationary. The driving factors of carbon emission intensity have obvious temporal and spatial heterogeneity. Therefore, we should fully consider the actual situation of each region and formulate differentiated emission reduction policies. For regions with high carbon emissions and low carbon emission intensity, such as Changsha and Changde, we should vigorously develop high-tech industries and promote technological innovation. For cities with high total and intensity of carbon emission, such as Xiangtan, Yueyang, and Loudi, we should actively change the energy structure and introduce modern service industries and high-end manufacturing industries to drive the transformation and upgrading of traditional industries.

However, there are still some limitations in this study. The reference value of carbon emission coefficient in this study comes from the existing research literature. Selecting the research results of Hunan Province and its surrounding similar areas, in order to reduce the negative impact of single error in the research results, the average value method is adopted, which has been reflected in other scholars’ research [34,37] and has certain rationality, but it is not as accurate as field sampling and survey [42,43]. In order to further improve the accuracy of the research results, the follow-up study should carry out an in-depth analysis of the carbon emission coefficient according to the actual situation of each city in Hunan Province, and consider the impact of inter-annual differences to obtain a more suitable carbon emission coefficient for each city in Hunan Province.

Due to the limitation of data, this paper only studies the changes in urban total carbon emissions in Hunan province from 2000 to 2020. The sample data need to be further enriched in the future, aiming to use smaller scale units, such as counties, in order to explore the spatiotemporal dynamic process of carbon emissions and its mechanism of action in depth. The GTWR model is used to analyze the influencing factors of carbon emissions. Actually, there are many driving factors that affect carbon emissions. This study does not decompose all the factors. It is hoped that future research can further comprehensively and deeply analyze the influencing factors of carbon emissions from land use. In addition, under the requirement of the “double carbon” goal, the research on carbon emissions should not be limited to the development status and the laws of temporal and spatial changes in the past, but should also be combined with LSTM, ARIMA-BP, and other models to carry out prediction and simulation research on carbon emissions and carbon emissions intensity at key time nodes in the future, which will be the focus of subsequent research.

## 6. Policy Suggestions

At present, Hunan Province is in a period of rapid industrialization and urbanization. With the advancement of urbanization, the area of cropland and forest will be reduced and the scale of construction land will be expanded. In addition, the continuous advancement of industrialization will also bring about a rapid increase in energy consumption, which will lead to a substantial increase in the carbon emissions of construction land. This paper studies the carbon emissions in Hunan Province from the perspective of spatiotemporal integration. Based on the above research conclusions, combined with the existing economic development level and resource endowment conditions of Hunan Province, the following suggestions are put forward, in order to provide a useful reference for Hunan Province to formulate “common but differentiated” carbon emission reduction policies.

(1) Optimize the land use structure and limit the excessive expansion of construction land. Construction land is the most important carbon source in Hunan Province, and it is the accumulation area of industry, construction, and transportation. It is also the key to reducing carbon emissions and achieving carbon emission reduction caused by the disorderly expansion of construction land. By adjusting and optimizing the internal structure and layout of construction land, the carbon pollution of high-carbon emission land can be reduced. Priority should be given to ensuring the land supply of emerging industrial projects with low energy consumption, low emissions, and high technology content, limiting and gradually reducing the land for traditional industrial projects with high energy consumption, high pollution, low efficiency, and explicitly those eliminated by the government, promoting the rationalization and low carbonization of land use layout.

(2) Increase ecological land and improve carbon sink capacity. Optimize the spatial pattern of land, build an ecological security pattern with Dongting Lake Wetland, Wuling Mountain Range, Luoxiao Mountain Range, and Nanling Mountain Range as ecological barriers, expand ecological land use, and focus on the construction of nature reserves, ecological function reserves, wetland reserves, and forest geological parks to maintain ecological security. Strictly control the ecological red line, protect biodiversity, improve ecological vitality, enhance the resilience of the ecosystem itself, and increase ecological land. Protecting the natural ecological environment of Dongting Lake is an important way to reduce carbon emissions in Hunan Province, as is implementing policies of “returning farmland to forest”, “returning farmland to lake”, and “returning farmland to grassland”. Increase the urban carbon sink function, increase urban green area, balance the network layout of urban green space, and strengthen the accessibility of green space. Protect the original urban water systems, lakes, wetlands, etc., and increase the carbon sink capacity of the urban water ecosystem.

(3) Optimize the industrial structure and fully tap the potential of energy conservation and emission reduction. The secondary industry, as a pillar industry in Hunan Province for a long time, is an important reason for its high average energy consumption level. Further promote the adjustment, optimization, and upgrading of the industrial structure, promote the transformation of the industrial structure from “two three one” to “three two one” mode in Hunan Province, and vigorously develop the tertiary industry with low emissions, low energy consumption, and high efficiency. At the same time, as Hunan Province is unlikely to surpass the economic development stage characterized by heavy chemical industry in a short time, the government should improve the entry threshold of energy conservation and environmental protection, control and gradually reduce the proportion of energy-consuming industries, and take the sustainable development path of “primary industry” stable development, “secondary industry” low-carbon development, and “tertiary industry” accelerated development.

(4) Improve the level of science and technology and energy efficiency, optimize the energy structure, and develop clean energy. The innovation and progress of science and technology is an important way to achieve energy conservation and emission reduction. Compared with developed countries, there is still a large space for improvement in the current energy utilization and technical level in Hunan Province. On the one hand, the government and enterprises should actively participate in the exchange and cooperation of international advanced technology and experience. Introduce, digest, and gradually develop energy-saving and emission-reduction technologies that can improve traditional utilization efficiency, promote cleaner production and green recycling development of enterprises, and provide strong technical support for Hunan Province to realize the decoupling of economic growth, energy consumption, and carbon emissions. On the other hand, according to the natural endowment of energy resources in Hunan Province, control and gradually reduce the share of coal in the energy consumption structure, improve and popularize clean coal production technology, increase the consumption proportion of high-quality energy such as natural gas with high unit calorific value, low carbon content, and high efficiency, and further develop and utilize clean energy such as solar energy, wind energy, biomass energy, water energy, and geothermal energy to form a low-carbon energy structure and realize coordinated and sustainable development of the economy, resources, and environment.

In view of the differences in carbon emissions among regions, we should establish a spatial concept, and on the basis of the construction of “3+5 urban agglomeration of Changsha-Zhuzhou-Xiangtan”, Changsha, with the highest carbon emission efficiency, should be used as the core city to radiate the whole province; Changde should be taken as the interface to improve the carbon emission efficiency in northwest Hunan; and Zhangjiajie should be linked with the low-carbon development in western and central Hunan. On the basis of promoting the formation of complementary space between southern and western Hunan, overall carbon emission efficiency will be accelerated from the edge to the central region. In terms of policy support, we should pay attention to improving the current situation of energy use in central Hunan and promoting the development of high-tech industries.

## Figures and Tables

**Figure 1 ijerph-20-03062-f001:**
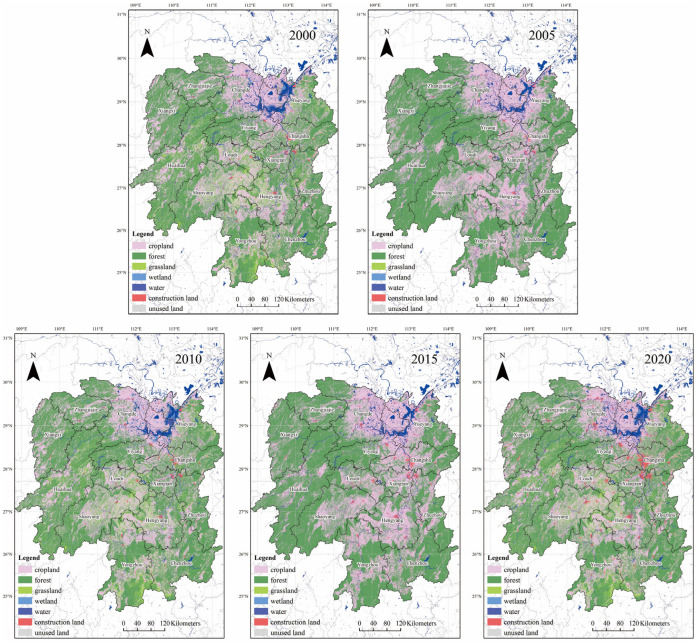
Administrative division and land use classification map of each city and prefecture in Hunan Province.

**Figure 2 ijerph-20-03062-f002:**
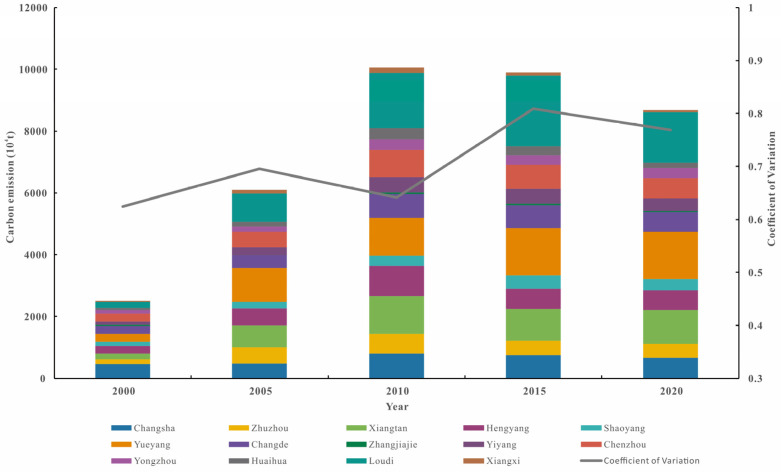
Characteristics of carbon emissions of each city in Hunan Province from 2000 to 2020.

**Figure 3 ijerph-20-03062-f003:**
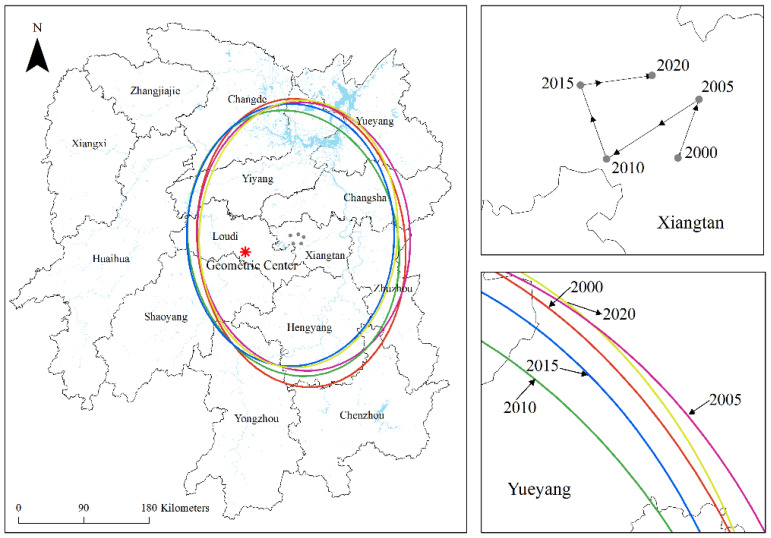
Ellipse distribution of carbon emission standard deviation and the movement trajectory of the center of gravity in Hunan Province.

**Figure 4 ijerph-20-03062-f004:**
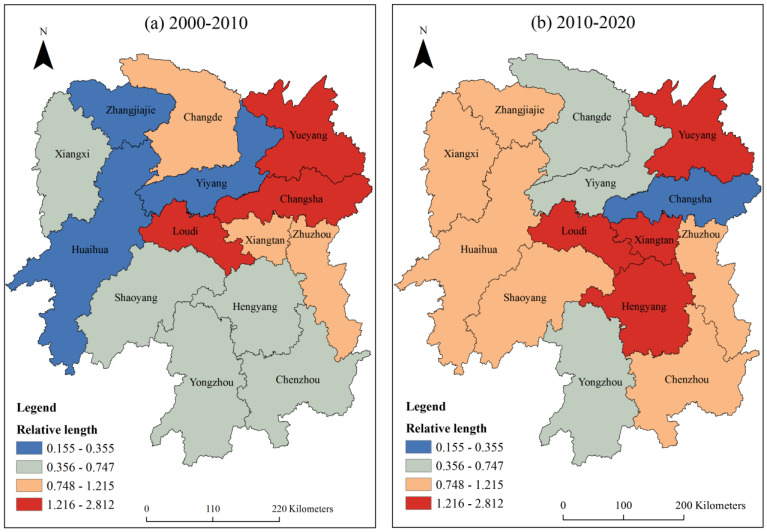
Spatial distribution of relative length of the LISA time path from 2000 to 2020.

**Figure 5 ijerph-20-03062-f005:**
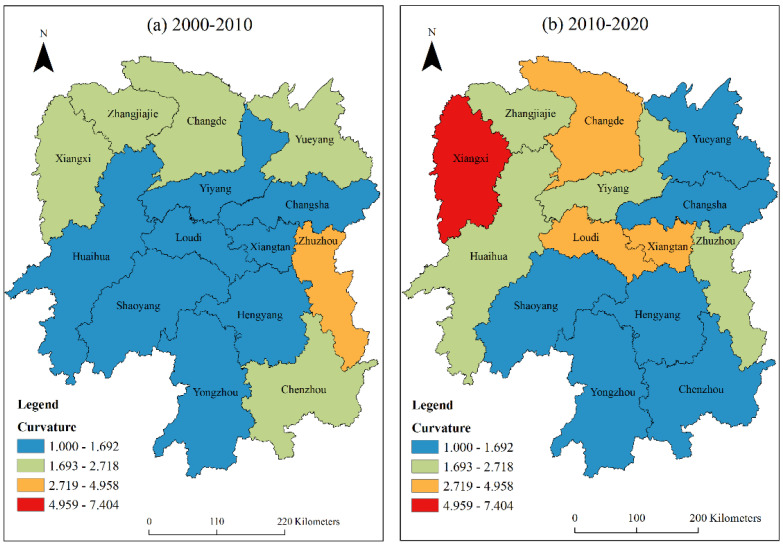
Spatial distribution of tortuosity of the LISA time path from 2000 to 2020.

**Figure 6 ijerph-20-03062-f006:**
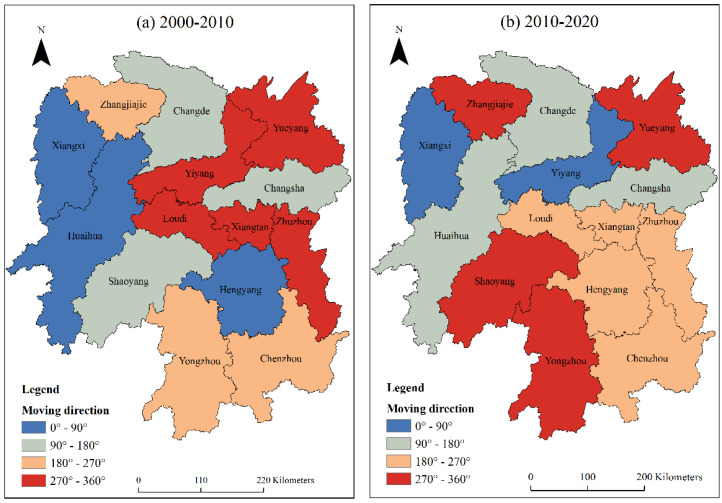
Spatial distribution of average movement direction of the LISA time path from 2000 to 2020.

**Figure 7 ijerph-20-03062-f007:**
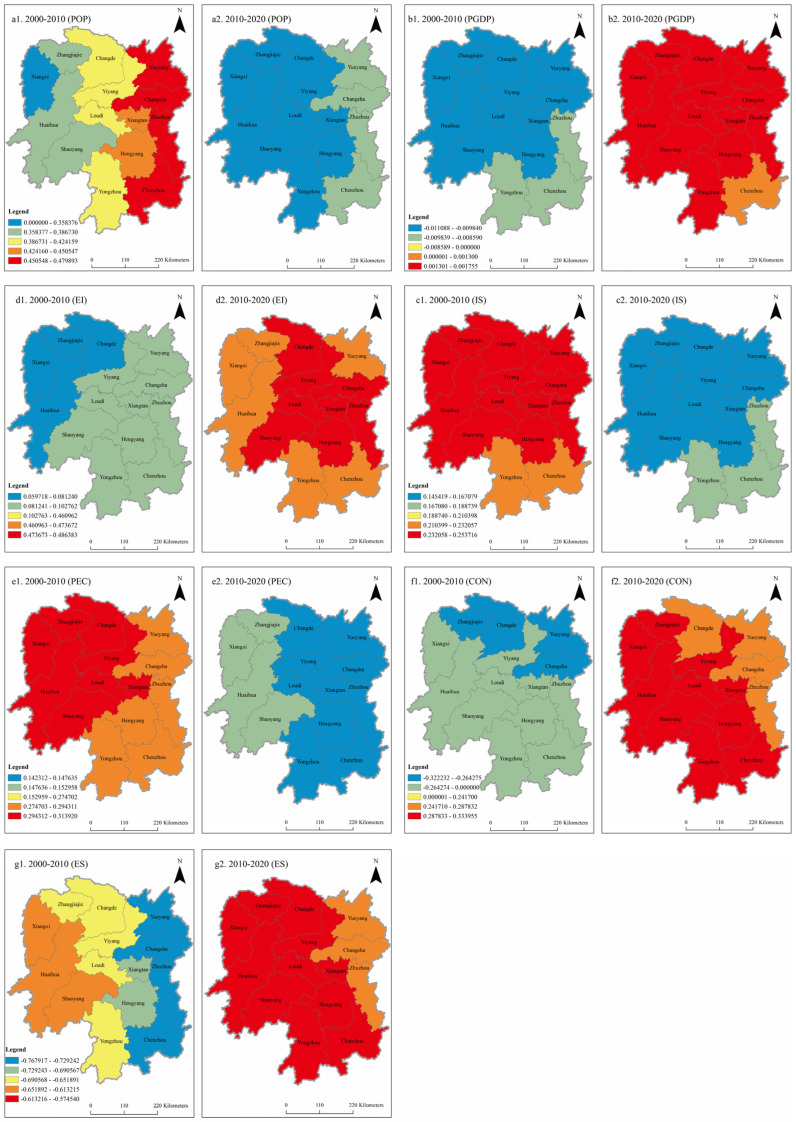
Spatial and temporal distribution of the influence of carbon emission in Hunan Province from 2000 to 2020.

**Table 1 ijerph-20-03062-t001:** Carbon emission coefficients and literature sources for each land use type.

Land Use Type	Carbon Emission Coefficients (t·hm^−2^)	Reference
cropland	0.497	[34,35,36]
forest	−0.581	[37,38]
grassland	−0.021	[39,40]
water	−0.253	[41,42]
unused land	−0.005	[27,40,43,44]

**Table 2 ijerph-20-03062-t002:** Standard coal conversion coefficient and carbon emission coefficient of various energy sources.

Energy Type	Standard Coal Conversion Coefficient	Carbon Emission Coefficient (in C, t·t^−1^)	Energy Type	Standard Coal Conversion Coefficient	Carbon Emission Coefficient (in C, t·t^−1^)	Energy Type	Standard Coal Conversion Coefficient	Carbon Emission Coefficient (in C, t·t^−1^)
Raw coal (kg·kg^−1^)	0.7143	0.7559	Other gas (kg·m^−3^)	0.1786	0.3548	Fuel oil (kg·kg^−1^)	1.4286	0.6185
Clean coal (kg·kg^−1^)	0.9000	0.7559	Clean Other coking products (kg·kg^−1^)	1.1000	0.6449	Liquefied petroleum gas (kg·kg^−1^)	1.7143	0.5042
Other washed coal (kg·kg^−1^)	0.2857	0.7559	Crude (kg·kg^−1^)	1.4286	0.5857	Other petroleum products (kg·kg^−1^)	1.4286	0.586
Coke (kg·kg^−1^)	0.9714	0.855	Gasoline (kg·kg^−1^)	1.4714	0.5538	natural gas (kg·m^−3^)	1.2143	0.4483
Coke over gas (kg·m^−3^)	0.5714	0.3548	Kerosene (kg·kg^−1^)	1.4714	0.5714	Heat power (kg·MJ^−1^)	0.0341	0.26
Blast furnace gas	0.1286	0.4602	Diesel oil (kg·kg^−1^)	1.4571	0.5921	Electricity (kg·(kW·h)^−1^)	0.1229	2.5255

**Table 3 ijerph-20-03062-t003:** Ellipse parameters of carbon emission standard deviation in Hunan Province.

Year	Coordinates of Gravity Center		Shift of Gravity Center			Long Axis/km	Minor Axis/km	Azimuth Angle/°
	Longitude	Latitude	Direction	Distance/km	Speed/km·a^−1^			
2000	112°23′59″ E	27°43′18″ N	——	——	——	143.01	199.24	173.39
2005	112°25′43″ E	27°47′36″ N	Northeast	9.55	1.91	146.50	185.12	175.96
2010	112°18′6″ E	27°43′13″ N	Southwest	16.85	3.37	144.06	184.62	168.33
2015	112°15′57″ E	27°48′38″ N	Northwest	11.99	2.4	142.84	180.76	0.23
2020	112°21′51″ E	27°49′21″ N	Northeast	11.05	2.21	136.58	184.39	3.74

**Table 4 ijerph-20-03062-t004:** Moran’s *I* Transition probability matrix and spatiotemporal transition of carbon emissions/%.

Period	t~t+1	HH	LH	LL	HL	Type	Number	Proportion	SF	SC
2000~2005	HH	50	0	0	50	I	7	50.00	50	50
	LH	33.33	33.33	33.33	0	II	2	14.29		
	LL	0	25	75	0	III	5	35.71		
	HL	33.33	0	33.33	33.33	IV	0	0.00		
2005~2010	HH	75	25	0	0	I	11	78.57	21.43	78.57
	LH	0	100	0	0	II	2	14.29		
	LL	0	20	60	20	III	1	7.14		
	HL	0	0	0	100	IV	0	0.00		
2010~2015	HH	66.67	33.33	0	0	I	12	85.71	14.28	85.71
	LH	0	75	25	0	II	1	7.14		
	LL	0	0	100	0	III	1	7.14		
	HL	0	0	0	100	IV	0	0.00		
2015~2020	HH	100	0	0	0	I	13	92.86	7.14	92.86
	LH	25	75	0	0	II	1	7.14		
	LL	0	0	100	0	III	0	0.00		
	HL	0	0	0	100	IV	0	0.00		

**Table 5 ijerph-20-03062-t005:** Definition and statistical description of related variables.

Variables	Description	Unit	Average	Standard Deviation	Minimum	Maximum	VIF
Population (POP)	Total population (T-POP)	Ten thousand people	476.00	174.66	147.81	1006.39	2.745
Economic development level (PGDP)	GDP per capita	RMB/person	28,022.08	26,053.41	2534	123,296.81	4.309
Industrial structure (IS)	Secondary industry/tertiated industry	%	100	36	23	178	1.456
Technological progress (EI)	Energy consumption per unit GDP	t/10,000RMB	0.94	0.64	0.04	2.88	2.739
Per capita energy consumption (PEC)	Per capita energy consumption	t/person	1.33	1.41	0.16	7.03	2.031
Land use (CON)	Built-up area	km^2^	288.35	155.82	34.86	932.88	5.460
Ecological environment (ES)	Forest coverage	%	60	10	40	77	1.206

**Table 6 ijerph-20-03062-t006:** Related parameters of GTWR.

Model Parameter	Bandwidth	Sigma	Residual Squares	AICc	R^2^	Adjusted R^2^	Spatiotemporal Distance Ratio
Value	0.150444	81.2702	462,339	954.456	0.969799	0.966389	4.89398

## Data Availability

The basic data used in the research can be found on the website of National Bureau of Statistics, Hunan Statistical Yearbook, and the Resource and Environmental Science and Data Center of the Chinese Academy of Sciences (http://www.resdc.cn (accessed on 20 August 2022)).

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
