# Peer review of "Spatiotemporal Dynamic Evolution and Its Driving Mechanism of Carbon Emissions in Hunan Province in the Last 20 Years"

_ijerph, 2023, doi:10.3390/ijerph20043062_

Round 1

Reviewer 1 Report

Report enclosed

Author Response

Responses to the comments of Reviewer (C, Comments; R, Response):

C: 1. Check the English

R: Thanks very much to the reviewer for this suggestion. We have modified English grammar and description.

C: 2. I suggest to explain the results better, that is in a comprehensive way. See for example the Abstract (from row 23 to row 38):

  1. there were 64% cities with a relative length (?) less than average length (?) My comment: what does that mean?
  2. The spatiotemporal agglomeration index has been greater than 50%, indicating that. have a certain stability. My comment: so what?
  3. The economic development level and the ecological environment have negative impacts on carbon emissions and the population, industrial structure, technological progress have positive impact on carbon emissions. This is counterintuitive: economic development and technological progress or industrial structure are usually moving together. I suspect some collinearity.

R: Thanks very much to the reviewer for this suggestion. We have added the result explanation. See Line 22-38. The variance expansion factor (VIF) of all variables is less than 10, which indicates that there is no multicollinearity in all normalized variables. The technical efficiency has not reached the forefront of production in high carbon emissions area in southern Hunan, and the central Hunan region has made great efforts in environmental governance, energy conservation and emission reduction, and the level of technological progress is far lower than that of other regions in the province, while the driving force of technological progress in western Hunan is insufficient, resulting in the low efficiency of technological progress in Hunan Province, thus the impact on carbon emissions is still positive. The contribution of the economic development level (PGDP) to carbon emissions has changed from negative to positive, and the absolute value of the regression coefficient has shown a decreasing trend, indicating that the impact of the PGDP on carbon emissions has gradually weakened. Since the improvement of the economic development level will lead to the transformation of the industrial structure, the economy has begun to develop in an intensive direction, the efficiency of resource utilization has been continuously improved, and people's awareness of environmental protection has been continuously improved. As a result, the intensity of carbon emissions continues to decline. Therefore, the main body of economic development level shows a negative impact. See Table 5, Line 467-470, 499-520, 521-539.

C: 3. 2.3 Name the subsection

All over the paper: There should be a specific Section for data (now a subsection 2.3.1) and a specific Section for methodology (now 2.3.2, 2.3.3, etc ) . For the methodology, I suggest to move definitions and description of the methodology to an Appendix. If in the main body, the reader is distracted by the formulas and the comments are brief and scattered now and then. Same for 3.3, 3.4, 3.5. The

plethora of applications and methods confuses the reader.

R: Thanks very much to the reviewer for this suggestion. We have put the method description of ESTDA and GTWR in the Appendix. See 2. Methodology and Data, Appendix.

C: 4. Give a clear and complete prospectus of the results.

The general feeling is, after all, what are the conclusions? Example: row 748: Among the moving directions types.. the spatial pattern change of carbon emissions in Hunan province has a certain spatial integration, but shows an increasing trend . My comment: so what?

R: Thanks very much to the reviewer for this suggestion. We have added in-depth discussion of the analysis results. See Line 640-718.

Reviewer 2 Report

i am honored to review this paper, i have following suggestions:

(1) In the introduction section, the literature review is insufficient and the main contribution of your research is not clearly defined in the theoretical sense.

(2)So many spatio-temporal analysis methods are used, why needs to explore the spatio-temporal association of carbon emission in Hunan, and the sample is less than 30, maybe not suitable for spatial analysis. besides, The migration of gravity center and SDE analysis of carbon emissions should be moved before the LISA analysis. 

(3)In terms of the part "Time evolution of driver influence", the influence analysis of each variables should be supported  by references.

(4) the spatial pattern varies different in Figure 7, please add more explanation, why select the 2010 as the time node.

(5)The managerial implications are not explained. besides,  Limited ideas for future research are offered based on the findings of the paper. How does your discussion section respond to previous research and provide insights for subsequent researches.

Author Response

Responses to the comments of Reviewer (C, Comments; R, Response):

C: 1. In the introduction section, the literature review is insufficient and the main contribution of your research is not clearly defined in the theoretical sense.

R: Thanks very much to the reviewer for this suggestion. We have revised the literature review to describe more relevant literature review. See Line 64-99.

C: 2. So many spatio-temporal analysis methods are used, why needs to explore the spatio-temporal association of carbon emission in Hunan, and the sample is less than 30, maybe not suitable for spatial analysis. Besides. The migration of gravity center and SDE analysis of carbon emissions should be moved before the LISA analysis.

R: Thanks very much to the reviewer for this suggestion. The migration of gravity center and SDE analysis of carbon emissions have been moved before the LISA analysis. See 3.3. The migration of gravity center and SDE analysis of carbon emissions. Compared with the spatial autocorrelation analysis method, the spatial autocorrelation analysis has been used in many studies, but the spatial autocorrelation analysis cannot further analyze the migration in time scales. The LISA time path is integrated to incorporate the dynamic migration of LISA coordinates in the Moran’s I scatter plots. Through the pairwise movement of the attribute and spatial lag values of the carbon footprint of prefecture-level cities over time, the spatial-temporal interactions and dynamic characteristics of the carbon footprint at the local level within regions are thus explained. While the spatial-temporal transition of LISA can reflect the transformation process of local spatial association types in the studied time range. And the spatial autocorrelation analysis and SBM-DEA model are relatively mature in carbon emissions research in Hunan Province. This paper wants to try a new analysis method to discuss the spatial and temporal pattern of carbon emissions in Hunan Province in the recent 20 years. The spatial pattern of carbon emissions is the result of the dynamic spatial effect of its own unit and its neighborhood unit. Even in the same province, there will be differences in carbon emissions of each city unit. Understanding the change rules of carbon emissions in different cities plays an important role in achieving the development of urban low-carbon economy and the regional "double carbon" goal. Therefore, the dynamic characteristics of spatial and temporal patterns of urban carbon emissions can be clearly discussed from this spatial scale. To sum up, it is necessary to analyze the dynamic changes of the spatial and temporal pattern of urban carbon emissions in central China from the city scale, which is the scientific basis for formulating differentiated urban emission reduction strategies. Therefore, this paper uses LISA method to analyze.

C: 3. In terms of the part "Time evolution of driver influence", the influence analysis of each variables should be supported by references.

R: Thanks very much to the reviewer for this suggestion. according to the research of other scholars on the driving factors of carbon emissions, the explanatory variables affecting carbon emissions include population, economic development level, industrial structure, technological progress, energy intensity, land use and ecological environment. We have added the references. See Line 462-466.

C: 4. the spatial pattern varies different in Figure 7, please add more explanation, why select the 2010 as the time node.

R: Thanks very much to the reviewer for this question. We have added more explanation of the GTWR results. See Line 480-624. Because the carbon emissions from 2000 to 2010 are a continuous rising stage, while a continuous declining stage from 2010 to 2020. Besides, In September 2009, the Chinese government formulated the carbon emission reduction policy, and since then, local governments have paid more and more attention to carbon emission reduction. Therefore, 2010 year could be taken as a node. See Line 329-333.

C: 5: The managerial implications are not explained. besides, Limited ideas for future research are offered based on the findings of the paper. How does your discussion section respond to previous research and provide insights for subsequent researches.

R: Thanks very much to the reviewer for this suggestion. We have added managerial implications and the inadequacies of the paper, and modified the Conclusion. See Line 640-718, 751-760.

Round 2

Reviewer 1 Report

Accepted with minor revision of English

Author Response

Responses to the comments of Reviewer (C, Comments; R, Response):

C: 1. Accepted with minor revision of English

R: Thanks very much to the reviewer for this suggestion. We have modified English grammar and description.

Reviewer 2 Report

1. The discussion section should be listed separately.

2. Limitations of the study should be supplemented.

3. Practical policy recommendations should be clearer.

Author Response

Responses to the comments of Reviewer (C, Comments; R, Response):

C: 1. The discussion section should be listed separately.

R: Thanks very much to the reviewer for this suggestion. The conclusion and discussion section have been listed separately and put the discussion section into the Policy suggestions section. See 6 Policy suggestions.

C: 2. Limitations of the study should be supplemented.

R: Thanks very much to the reviewer for this suggestion. The limitations of this study have been described in the conclusion. See Line 734-757.

C: 3. Practical policy recommendations should be clearer.

R: Thanks very much to the reviewer for this suggestion. We have added in-depth discussion of the analysis of the practical policy recommendations. See 6 Policy suggestions.

Round 3

Reviewer 2 Report

Great improvemetns.